# TWO-BRANCH LABEL DISTRIBUTION LEARNING

## ABSTRACT

Label Distribution Learning (LDL) is a novel machine learning paradigm that assigns label distribution to each instance, capturing comprehensive message by predicting description degree. However, existing approaches have primarily focused on single label distribution settings, limiting their applicability to tasks with only one output branch and rendering them sensitive to numerical fluctuations. For the first time, this paper introduces the problem of Two-Branch Label Distribution Learning (TBLDL), where an LDL model is supervised by two distinct label distributions. We assume that there is a correlation between two label branches, and by modeling this interdependence, we can effectively fuse cross-branch information to enhance predictive performance. Furthermore, for conventional LDL tasks involving only a single label distribution, our framework enables the generation of an auxiliary smooth distribution, thereby improving model robustness and accuracy. Finally, we apply TBLDL into several traditional LDL methods, demonstrating its broad compatibility and generalizability. Experimental results confirm that TBLDL not only handles both one and two-branch LDL tasks effectively but also consistently outperforms traditional LDL approaches.

## 1 INTRODUCTION

Label distribution learning (LDL) Geng (2016) is a novel learning paradigm, mainly focuses on the ambiguity at label side, i.e., one instance is not necessarily mapped to one label, but assigns with a label distribution. As shown in Fig. 1, a label distribution is a multidimensional vector describes the relative importance of labels, in where each element is called the label description degree. Label distribution has been widely used in a variety of fields, such as age estimation Geng et al. (2013), emotion recognition Jia et al. (2019), and acne severity prediction Wu et al. (2019).

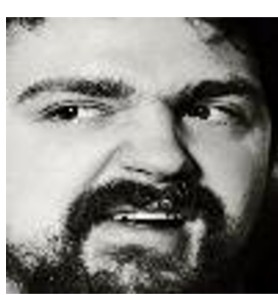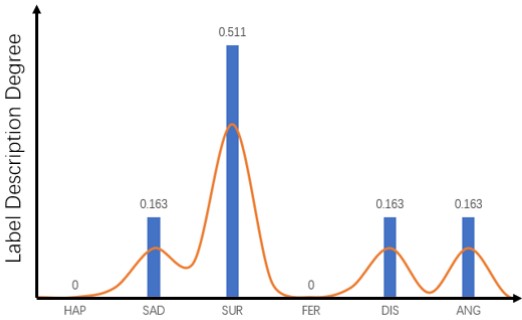

Figure 1: Example of label distribution. The image is assigned with output layer label description degrees of 0.163, 0.163, 0.163 and 0.511 for "SAD,""DIS,""ANG," and "SUR" respectively, and 0's for other labels ("HAP," "SAD," "SUR," "FER," "DIS," and "ANG" represent "Happy," "Sad," "Surprised," "Fear," "Disappointed," and "Angry").

LDL is helpful to handle label ambiguity He et al. (2016). It is able to learn the intrinsic links between labels as opposed to only predicting a single label value. For example, in real-world difficulties, people tend to estimate the grade based on an approximate value, without needing a precise value (e.g., this person has around 10 acne and is slightly symptomatic; another person has roughly 50 acne and is extremely symptomatic). Wu et al. (2019) uses LDL to learn to predict a Gaussian

distribution with a peak at the true point, and then sums the confidence regions of the regions corresponding to different grades to predict the grade, tends to work better than predicting the grade directly, and is in line with common human cognitive habits.

In previous study Wu et al. (2019), the mapping between acne severity and lesion counts is directly given a priori called Hayashi criterion Wu et al. (2019). We aim to refine it by learning that information through subsequent procedures, getting rid of the model's reliance on fixed priori knowledge.

We propose a new LDL algorithm called TBLDL, which automatically learns the relationships among label branches rather than relying on predefined or fixed mappings. Our main assumption is that there is a correlation between different label branches, and automatically detecting it helps in inference. First, we treat the labels from different branches within the dataset as distinct label distributions; for datasets containing only a single label branch, an auxiliary hidden layer is automatically introduced (indicated by the orange line). Second, a transfer matrix that characterizes the inter-branch mapping is learned through a Markov process during the iterative training procedure. Finally, information from the hidden layer is projected onto the output layer. Experimental results demonstrate that TBLDL achieves superior performance compared to methods that require explicit specification of label relationships. In addition, we introduce our framework into some traditional LDL methods. By using two label distributions and learnable transform matrix, the performance of these traditional methods can been improved, illustrating the universality and flexibility of TBLDL.

Our contributions are as follows:

- For issues that already include two label distributions, such as the rank assessment problem, our approach is able to learn the intrinsic dimensionality between the data and get rid of the dependency on fixed priori knowledge.
- For most of the issues comprising only one label distribution branch, we may allow the model to learn more fine-grained knowledge and produce better prediction results by generating a hidden dimension.
- Introducing TBLDL into the traditional LDL method can lead to performance improvement, illustrating the broad extensibility and research value of our framework.

## 2 RELATED WORK

As a revolutionary learning paradigm, LDL integrates ambiguity in labels called label distribution to precisely capture the degree of labelling of each instance, instead of typical multi-label learning. This particular characteristic has inspired tremendous scientific interests in the realm of LDL. In this section, we provide a brief introduction to existing studies. The existing LDL algorithms can be broadly divided into three groups. The first group turns LDL problem into a single-label learning problem by sampling from the raw LDL data using the description degree of that label as sample weight. There are two main algorithms represented within this group: PT-SVM and PT-Bayes Geng (2016), which utilize the SVM algorithm and the Bayes classifier to solve the transformed weighted single-label learning problem. The second group is based on algorithm adaption, which employs machine learning algorithms to deal with LDL problem. For example, the K-nearest neighbors (KNN) classifier finds the top k neighbors for an instance and uses their average labels as the prediction of that instance. Backpropagation (BP) neural networks directly outputs the descriptive degree. The final group of LDL algorithms are those comprises specialized algorithms such as IIS-LDL and BFGS-LDL, whose objective is to optimizing the sum of log-likelihood of labels and instances. They consider LDL as a regression problem and utilize improved iterative scaling and quasi-Newton methods to solve it.

Our work is different from the previous studies as follows: First, the existing LDL algorithms studied on only one label branch. On the contrary, we are the first to consider issues having two branches of labels. By focusing on understanding the connections between two label branches, we extract the knowledge of the hidden layer to output layer and gain better results. Second, for problems only containing one output layer, we move the original problem to our Two-layer LDL problem by generating a hidden layer. The smoother hidden layer contributes to better performance. Finally, we extend our framework to traditional LDL methods, showing the generality of our framework. In summary, we offer a new LDL framework that is capable of detecting connections between distinct label distribution branches and can be extend to standard LDL approaches.

## 3 METHOD

### 3.1 FORMALISATION

First, we introduce the main notations used in this paper. Instance variables are denoted by $x$, and a specific instance is represented as $x_i$. Each instance $x_i$ is associated with two label distributions: the hidden layer distribution $d_i$ and the output layer distribution $l_i$, defined as follows:

$$l_i = (d_{x_i}^{y_1}, d_{x_i}^{y_2}, \ldots, d_{x_i}^{y_c}) \in \mathbb{R}^c \tag{1}$$

$$d_i = (d_{x_i}^{z_{11}}, \ldots, d_{x_i}^{z_{1n_1}}, \ldots, d_{x_i}^{z_{c1}}, \ldots, d_{x_i}^{z_{cn_c}}) \in \mathbb{R}^d, \tag{2}$$

where $c$ denotes the dimension of output layer and $d$ denotes the dimension of hidden layer. The output layer description degree $d_{x_i}^{y_k}$ can be expressed as the sum of corresponding hidden layer description degrees:

$$\sum_{j=1}^{n_k} d_{x_i}^{z_{kj}} = \sum_{z \in \phi(k)} d_{x_i}^z = d_{x_i}^{y_k}. \tag{3}$$

Here, $\phi(k) = \{z_{k1}, z_{k2}, \ldots, z_{kn_k}\}$ represents the set of hidden units associated with output label $y_k$, and $n_k$ denotes the cardinality of $\phi(k)$.

The method chapter is organized into three sequential steps. First, for datasets that already contain two output branches, we describe how to transform the output branches into label distributions and illustrate the operation of our TBLDL framework. Second, for datasets with only a single output branch—where only the output layer distribution $L$ is available—we present a method to generate the corresponding hidden layer distribution $D$, thereby reformulating the original single-branch problem as a TBLDL task. Finally, we integrate TBLDL paradigm into several conventional label learning approaches, demonstrating consistent performance improvements across multiple settings.

### 3.2 PROBLEM WITH TWO LABEL BRANCHES

The ACNE04 dataset contains two label branches: acne count and acne severity. Given $n$ input training images $x_i$, each associated with lesion count $z_i$ and acne severity level $y_i$: $\{(x_1, y_1, z_1), \cdots, (x_n, y_n, z_n)\}$. We treat lesion counts as hidden layer and acne severity as output layer, taking advantage of the hidden layer to infer the output layer by using TBLDL framework.

#### 3.2.1 GENERATING LABEL DISTRIBUTIONS

To exploit TBLDL framework, we first transform the scalar-valued labels into label distributions. For the severity labels $y_k$, they are encoded as one-hot vectors; for the counting labels $z_j$, a Gaussian function is employed following Gao et al. (2017). The description degree of a specific acne lesion count label $z_j$ with respect to instance $x_i$ is defined as:

$$d_{x_i}^{z_j} = \frac{1}{\sqrt{2\pi}\sigma M} \exp(-\frac{(z_j - z_i)}{2\sigma^2}), \tag{4}$$

where $z_i$ denotes the true lesion count of instance $x_i$, and the standard deviation $\sigma$ controls the spread of the distribution. As in Wu et al. (2019), we set $\sigma = 3$. The normalization factor $M = \frac{1}{\sqrt{2\pi}\sigma} \sum_{z \in \phi(k)} \exp(-\frac{(z - z_{\text{center}})}{2\sigma^2})$ ensures the resulting label distribution satisfies the probabilistic constraints: $d_{x_i}^{z_j} \in [0, 1]$ and $\sum_{j=1}^d d_{x_i}^{z_j} = 1$.

#### 3.2.2 HIDDEN LAYER

To leverage the hidden layer for inferring the output layer, we first predict the hidden layer label distribution—specifically, the lesion count. For an input image $x_i$, it is resized and passed through a ResNet-50 He et al. (2016), and a multilayer perceptron(MLP) Hornik et al. (1989) maps the extracted CNN features to the output space of hidden layer. Finally, we use Softmax to normalize:

$$p_i^{cnt} = Softmax(MLP_{cnt}(CNN(x_i))) \tag{5}$$

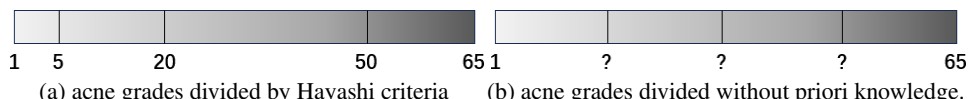

(a) acne grades divided by Hayashi criteria    (b) acne grades divided without priori knowledge.

Figure 2: Division of acne grades. (a) Hayashi criteria divided the acne into four grades. (b) Our model learns the mapping from lesion numbers to severity levels.

We employ the Kullback–Leibler (KL) divergence loss He et al. (2016) to assess the distance between the predicted distribution $p_i^{cnt} = [p_i^{(1)}, \cdots, p_i^{(d)}]$ and the true distribution $d_i = [d_{x_i}^{z_1}, \cdots, d_{x_i}^{z_d}]$:

$$L_{\text{cnt}}(d_i, p_i^{cnt}) = \sum_{j=1}^{d} (d_{x_i}^{z_j} \ln p_i^{(j)}). \tag{6}$$

### 3.2.3 TRANSFER HIDDEN LAYER TO OUTPUT LAYER

The second step in leveraging the hidden layer involves mapping it onto the output layer. Wu Wu et al. (2019) first predicts the lesion count distribution and defines the sum of lesion count label descriptors within each corresponding confidence interval $\phi(k)$ as the grading descriptor, based on the Hayashi criterion Hayashi et al. (2008), which is a clinical guideline used by dermatologists to assess acne severity by integrating lesion counts with global visual evaluation.

Inspired by this approach, we aim to avoid prescribing the mapping from lesion counts (hidden layer) to severity grades (output layer) through fixed, a priori rules such as the Hayashi criterion. Instead, we propose to learn this mapping by capturing the implicit relationships between label branches. As illustrated in Fig. 2, we model the transformation using a learnable transfer matrix $O$ rather than relying on predefined criteria. The transfer matrix $O$ is a binary 0–1 matrix that aggregates description degrees from intervals in the hidden layer to the output layer. Only the interval boundaries $n_1, n_2, ...$ are treated as learnable parameters; thus, $O$ can be expressed as $O_{0,n_1,n_2,\cdots,d}$.

We employ a Markov process to optimize the interval boundaries. Initially, the confidence intervals are set uniformly. For instance, in the acne grading task, the initial boundaries are set at 17, 33, and 49, partitioning the range [0, 65] into four equal subintervals.

During optimization, the goal is to update $O$ such that the transformed predicted counting distribution $p_i^{cnt} = [p_i^{(1)}, \cdots, p_i^{(d)}]$ better approximates the target severity distribution $l_i = [d_{x_i}^{y_1}, \cdots, d_{x_i}^{y_c}]$. Specifically, we minimize the discrepancy $\|p_i{}^{cnt}O - l_i\|$. In each iteration, the interval boundaries are adjusted based on the magnitude of this error, which serves as the basis for computing transition probabilities in the Markov process.

Let $q_i^{cntcls} = p_i^{cnt}O$ denote the prediction transferred from the hidden layer to the output layer. We compute the loss between the predicted distribution $q_i^{cntcls}$ and the true grading label distribution $l_i$:

$$L_{\text{cnt2cls}}(l_i, q_i^{cntcls}) = \sum_{j=1}^{d} (d_{x_i}^{y_j} \ln q_i^{(j)}). \tag{7}$$

### 3.2.4 OUTPUT LAYER

The final step involves directly predicting the output layer distribution, as the Hayashi criterion requires acne severity grading to be determined through a combination of global and local diagnostic assessments. Analogous to the modeling of the hidden layer distribution, the output layer distribution is generated directly via:

$$p_i^{cls} = Softmax(MLP_{cls}(CNN(x_i))), \tag{8}$$

and the KL loss Gao et al. (2017) is employed to measure the discrepancy between the predicted output label distribution $q_i^{cls}$ and the true output label distribution $l_i$:

$$L_{\text{cls}}(l_i, q_i^{cls}) = \sum_{j=1}^{d} (d_{x_i}^{y_j} \ln q_i^{(j)}). \tag{9}$$

### 3.2.5 MULTITASKING

The multiple loss-guided components discussed above focus on distinct aspects of the task. For a given instance $x_i$, the grading task performs a global assessment; the counting task emphasizes extracting local information related to individual lesions; and the transfer task aims to map such local details into the global representation, thereby leveraging hidden layer information to support the inference of the output layer. Our model integrates both global and local features, and the overall training loss is defined as:

$$L_i(x_i, y_i, z_i) = \frac{\lambda}{2}(L_{cnt}(d_i, p_i^{cnt}) + L_{cnt2cls}(l_i, q_i^{cnt})) + (1 - \lambda)L_{cls}(l_i, q_i^{cls}), \tag{10}$$

in where parameter $\lambda$ balances the contribution of the counting and grading tasks

### 3.3 PROBLEM WITH ONE LABEL BRANCH

Unlike the ACNE04 dataset, which contains two label branches, most datasets provide only a single label branch—namely, the output layer with dimension $c$. Inspired by this acne severity prediction framework, we propose to generate a hidden layer from the output layer, thereby transforming a single-label learning problem into a TBLDL-compatible task.

The method for generating the hidden layer is as follows. As illustrated in Fig.1, our goal is to discretize the description degree $d_{x_i}^{y_k}$ (represented by blue bars) over the confidence interval $\phi(k)$(indicated by the red line). We first partition each confidence interval $\phi(k)$ based on the observed minimum and maximum values within the dataset. Then, we distribute $d_{x_i}^{y_k}$ across the subintervals corresponding to lesion count values $z$ associated with class $y_k$, such that the aggregated sum satisfies $\sum\limits_{d_{x_i}^z \in \phi(k)} d_{x_i}^z = d_{x_i}^{y_k}$. Further, we require that $\sum\limits_{d_{x_i}^z \in \phi(k)} d_{x_i}^z$ obeys a Gaussian distribution:

$$d_{x_i}^z = \frac{d_{x_i}^{y_k}}{\sqrt{2\pi}\sigma M} \exp(-\frac{(z - z_{center})}{2\sigma^2}) \tag{11}$$

where $z_{center} = \frac{1}{2}(z_{k1} + z_{kn_k})$ denotes the midpoint of the confidence interval $\phi(k)$, and $M = \frac{1}{\sqrt{2\pi}\sigma} \sum\limits_{z \in \phi(k)} \exp(-\frac{(z-z_{center})}{2\sigma^2})$ serves as the normalization factor.

After generating the hidden layer, we apply the TBLDL framework to address the single-label problem. Experimental results show that TBLDL outperforms the original method, likely because the additional hidden layer incorporates statistical prior knowledge of category occurrence probabilities in the dataset.

### 3.4 APPLY TBLDL TO TRADITIONAL LDL

In the previous section, the mapping from features to labels is performed using an MLP. In this section, we replace the MLP with a traditional learning approach to demonstrate that our framework can serve as a general paradigm, whose generality and extensibility offer valuable potential for future research. Specifically, we generate the hidden layer label distribution $d_i$ using the method described in Section 3.3; then, we map the features to the output spaces of both the hidden and output layers using traditional label distribution learning (LDL) instead of MLP. Finally, the transfer matrix is learned through the Markov process. The complete algorithm is presented in Algorithm 1.

---

**Algorithm 1** algorithm of 4.5

---

**Input:** input feature $X$, output layer label distribution $L$, transfer matrix $O$, LDL method $method$
1: generate hidden layer distribution $D$ by weighted allocation method
2: **while** $stopping\ critertion\ is\ not\ satisfied$ **do**
3:    $D^{predict} = method(X, D)$
4:    $L^{predict} = method(X, L)$
5:    $O^{new} = markov(O, D^{predict}, L^{predict})$
6: **end while**

---

# 4 EXPERIMENTS

## 4.1 DATASETS & EVALUATION METRICS

### 4.1.1 DATASETS

In this section, we present a total of 13 datasets, with their detailed information summarized in Table 1 and Table 5. These datasets span a diverse range of domains, including biology, movie analysis, and facial expression recognition. This broad domain coverage enables a comprehensive evaluation of the adaptability of our proposed method across varied application contexts.

We categorize the datasets listed in Table 1 into three groups according to example dimension: datasets 1–6 are classified as small-scale, 7–10 as medium-scale, and 11–12 as large-scale.

| Index | Datasets | examples | features | labels |
|-------|----------|----------|----------|--------|
| 1 | Yeast_alpha | 2465 | 24 | 15 |
| 2 | Yeast_cdc | 2465 | 24 | 18 |
| 3 | Yeast_dtt | 2465 | 24 | 4 |
| 4 | Yeast_heat | 2465 | 24 | 6 |
| 5 | Yeast_spo | 2465 | 24 | 6 |
| 6 | Yeast_spo5 | 2465 | 24 | 3 |
| 7 | SJAFFE | 213 | 243 | 6 |
| 8 | SBU_3DFE | 2500 | 243 | 6 |
| 9 | Nature_Scene | 2000 | 294 | 9 |
| 10 | ffb5500 | 5500 | 512 | 5 |
| 11 | RAFML | 4908 | 200 | 6 |
| 12 | Movie | 7755 | 1869 | 5 |

Table 1: Statistics of the 12 data sets.

The ACNE04 dataset is presented separately in Table 5 and contains two label branches: Lesion, indicating the number of acne lesions, and Class, representing the severity grade. For evaluation, the dataset is partitioned into an 80% training set and a 20% test set.

### 4.1.2 EVALUATION METRICS

Geng Geng (2016) proposed six metrics to evaluate the performance of LDL algorithms. The formulas of these metrics are provided in Table 2. In these formulas, $p$ denotes the actual label distribution, $q$ denotes the predicted label distribution, $q_i$ denotes the $i$th element in $q$. "↓" indicates that smaller values are better, while "↑" indicates that larger values are preferred.

| LDL Metric | Formula |
|------------|---------|
| Chebyshev distance(Chebyshev)↓ | $\max_i \lvert p_i - q_i \rvert$ |
| Clark distance(Clark) ↓ | $\sqrt{\sum_i (p_i - q_i)^2 / (p_i + q_i)^2}$ |
| Canberra distance(Chebyshev) ↓ | $\sum_i \lvert p_i - q_i \rvert / p_i + q_i$ |
| KL divergence(KL) ↓ | $\sum_i p_i \ln \frac{p_i}{q_i}$ |
| Cosine similarity(Cosine) ↑ | $\frac{p^\top q}{\lVert p \rVert_2 \lVert q \rVert_2}$ |
| Intersection similarity(Intersection) ↑ | $\sum_i \min (p_i, q_i)$ |

Table 2: Evaluation Metrics for LDL.

## 4.2 PROBLEM WITH TWO LABEL BRANCHES

The experiments in this section correspond to Section 3.2. We use the ACNE04 dataset from Table 5 for training. Most of our settings follow Wu et al. (2019): we employ ResNet-50 He et al. (2016), pre-trained on ImageNet Russakovsky et al. (2015), as the feature extractor; input images are resized to $224 \times 224 \times 3$ pixels and normalized to $[0, 1]$ before training; we use Stochastic Gradient Descent (SGD) as the optimizer and train for 100 epochs, with momentum set to 0.8 and weight decay to

5e-4. The initial learning rate is 0.001, decaying by a factor of 0.5 every 25 epochs. All experiments are conducted on an NVIDIA GeForce RTX 4090 GPU with 24GB VRAM. For the hyperparameter $\lambda$, we retain the same value as in the original work, $\lambda = 0.6$. Notably, we increase the batch size from 32 to 64, because the update of the transition matrix depends on all instances within a batch, and a larger batch size improves the statistical reliability of the voting process.

We evaluate performance using grading accuracy on the test set, comparing direct incorporation of prior knowledge via the Hayashi criterion with learning the mapping through a transformation matrix. Results are reported in Table 3. These results motivate further investigation: the inclusion of a hidden layer enables the model to capture additional latent knowledge. In the subsequent experiments on datasets with only a single label branch, we explore automatically constructing the hidden layer label distribution from the output layer labels, thereby enabling training within the proposed TBLDL framework.

|  | Accuracy |
|---|---|
| Hayashi Criteration | 0.821 |
| Transform Matrix | **0.837** |

Table 3: Acne Severity Comparison on ACNE04 dataset

## 4.3 One Label Problem

The experiments in this section correspond to Section 3.3 and use the datasets listed in Table 1, all of which have a single label branch. Here, the mapping from features to outputs is learned using an MLP implemented on GPUs with sufficient computational capacity, allowing us to focus on larger-scale datasets. Specifically, we select two small datasets (1–2), four medium datasets (7–10), and two large datasets (11–12). Since all these datasets contain only one label layer, the first step is to generate the hidden layer label distribution. For each category $k$, we propose two interval division methods for determining the length of the confidence interval $n_k$: the first is the average division method, which uniformly allocates the hidden layer dimensions across categories, i.e., $n_k = w \times d$; the second is the weighted division method, where $w_k = \frac{\sum_{i=1}^{n} d_{x_i}^{y_k}}{\sum_{k=1}^{c} \sum_{i=1}^{n} d_{x_i}^{y_k}}$. A higher $w_k$ indicates a greater occurrence frequency of category $k$ in the dataset. For such categories, we assign longer confidence intervals: $n_k = w_k \times d$

The experiments are divided into three parts. First, we compare the performance of the two interval division methods, and the results show that the weighted method performs better. These results are presented in Table 6. Additionally, we adopt several state-of-the-art (SOTA) LDL algorithms as baselines, and their performance comparison is shown in Table 4. All SOTA methods use the parameter settings recommended in their original studies.

- AA-BP Geng (2016): AA-BP is a structure with a three-layer network. The network outputs different units, and each output unit represents the descriptive degree.
- AA-KNN Geng (2016): For each instance, AA-KNN first find its k nearest neighbors in the training set. Then, calculate the mean of the label distribution of all k nearest neighbours as the label distribution.
- PT-Bayes Geng (2016): PT-Bayes transforms the LDL problem into a single-label learning problem, then utilizes the Bayes classifier to address the transformed weighted single-label learning problem.
- LCLR Ren et al. (2019b): LCLR reconstructs a new supervised label distribution with global and local label-related information.
- LDLSF Ren et al. (2019a): LDLSF uses label-specific features to improve performance.
- LDLLC Jia et al. (2018): LDLLC utilizes local label correlation to make prediction between similar instance.
- CPNN Geng et al. (2013): Conditional Probability Neural Network, employs a three-layer neural network structure to learn the distribution of labels.

| | | chebyshev | clark | canberra | kldist | cosine | intersection |
|---|---|---|---|---|---|---|---|
| alpha | AA-BP | 0.0434±.0001 | 0.9215±.0005 | 3.1670±.0014 | 0.0885±.0093 | 0.9370±.0008 | 0.8393±.0002 |
| | AA-KNN | 0.0275±.0401 | 0.5602±.0232 | 1.9409±.0964 | 0.0336±.0349 | 0.9691±.0252 | 0.8955±.0811 |
| | CPNN | 0.0724±.0025 | 0.8629±.0018 | 3.2067±.0088 | 0.0933±.0045 | 0.9028±.0016 | 0.8121±.0015 |
| | LDSVR | 0.0294±.0029 | 0.5970±.0041 | 1.9463±.0157 | 0.0390±.0096 | 0.9652±.0028 | 0.8947±.0021 |
| | LCLR | 0.0151±.0016 | 0.2412±.0011 | 0.7946±.0031 | 0.0070±.0036 | 0.9930±.0013 | 0.9560±.0002 |
| | LDLSF | 0.0151±.0023 | 0.2413±.0032 | 0.7953±.0109 | 0.0070±.0070 | 0.9930±.0093 | 0.9560±.0032 |
| | LDLLC | 0.0145±.0029 | 0.2252±.0033 | 0.7344±.0083 | 0.4233±.0054 | 0.9939±.0019 | 0.9584±.0018 |
| | PT-Bayes | 0.3875±.0015 | 2.0231±.0010 | 7.6577±.0039 | 0.6384±.0048 | 0.4843±.0016 | 0.5185±.0009 |
| | OURS | **0.0134±0.0004** | **0.2115±0.003** | **0.686±0.0097** | **0.0019±0.0002** | **0.9946±0.0002** | **0.9639±0.0006** |
| cdc | AA-BP | 0.0669±.0024 | 1.0440±.0036 | 2.5764±.0079 | 0.1639±.0069 | 0.9139±.0021 | 0.8433±.0017 |
| | AA-KNN | 0.0448±.0106 | 0.6288±.0276 | 2.0304±.0790 | 0.0543±.0199 | 0.9484±.0213 | 0.8629±.0130 |
| | CPNN | 0.0620±.0058 | 1.4701±.0024 | 4.6585±.0080 | 0.2909±.0207 | 0.8524±.0011 | 0.7283±.0013 |
| | LDSVR | 0.0232±.0020 | 0.2772±.0036 | 0.8296±.0144 | 0.0094±.0091 | 0.9910±.0038 | 0.9459±.0020 |
| | LCLR | 0.0168±.0020 | 0.2237±.0027 | 0.6752±.0085 | 0.0074±.0012 | 0.9928±.0011 | 0.9555±.0011 |
| | LDLSF | 0.0168±.0020 | 0.2236±.0036 | 0.6748±.0144 | 0.0074±.0091 | 0.9928±.0038 | 0.9555±.0020 |
| | LDLLC | 0.0164±.0024 | 0.2147±.0036 | **0.6435±.0079** | 0.9225±.0069 | 0.9933±.0021 | 0.9576±.0017 |
| | PT-Bayes | 0.2408±.0042 | 1.9271±.0135 | 6.8458±.0342 | 0.5819±.0023 | 0.6639±.0026 | 0.5617±.0044 |
| | OURS | **0.0162±0.0003** | 0.217±0.0055 | 0.6496±0.0214 | **0.0041±0.0004** | **0.9933±0.0004** | **0.9588±0.0014** |
| sja | AA-BP | 0.4391±.0001 | 1.3493±.0004 | 2.7821±.0011 | **0.7564±.4300** | 0.5295±.0004 | 0.5365±.0006 |
| | AA-KNN | 0.1646±.0001 | 0.5756±.0005 | 1.0271±.0014 | 0.1006±.0093 | 0.9024±.0008 | 0.8354±.0008 |
| | CPNN | 0.1986±.0040 | 0.7276±.0040 | 1.3999±.0090 | 0.1796±.0085 | 0.8364±.0035 | 0.7681±.0021 |
| | LDSVR | 0.1716±.0795 | 0.6218±.0031 | 1.1184±.0042 | 0.1175±.0061 | 0.8917±.0016 | 0.8284±.0015 |
| | LCLR | 0.1277±.0039 | 0.4430±0057 | 0.9331±.0080 | 0.0792±.0018 | 0.9248±.0016 | 0.8400±.0759 |
| | LDLSF | 0.1240±.0051 | 0.4693±.0112 | 0.9814±.0232 | 0.0754±.0039 | **0.9309±.0037** | 0.8381±.0043 |
| | LDLLC | **0.1129±.0227** | 0.4690±.0731 | 0.9822±.0878 | 4.4745±.0565 | 0.9308±.0357 | 0.8380±.0256 |
| | PT-Bayes | 0.2714±.0584 | 0.7375±.0918 | 1.4657±.0192 | 0.2256±.0168 | 0.7842±.0190 | 0.7286±.0557 |
| | OURS | 0.1234±.0086 | **0.4374±0.0173** | **0.9199±0.0538** | 0.0761±0.0095 | 0.9276±0.0084 | **0.844±0.0105** |
| SBU | AA-BP | 0.3012±.0040 | 1.1851±.0080 | 2.7515±.0187 | 0.6154±1.5501 | 0.5961±.0037 | 0.5337±.0036 |
| | AA-KNN | 0.2419±.0029 | 0.6145±.0033 | 1.3260±.0083 | 0.3355±.0054 | 0.8456±.0019 | 0.6884±.0018 |
| | CPNN | 0.2689±.0053 | 0.8444±.0277 | 1.9320±.0656 | 0.3131±.0149 | 0.7359±.0077 | 0.6558±.0089 |
| | LDSVR | 0.2324±.0279 | 1.1012±.0238 | 2.4310±.0621 | 0.4319±.1620 | 0.7189±.0093 | 0.6033±.0076 |
| | LCLR | 0.1345±.0027 | 0.4134±.0068 | 0.9043±.0160 | 0.0848±.0021 | 0.9179±.0022 | 0.8384±.0028 |
| | LDLSF | 0.1383±.0000 | 0.4112±.0008 | 0.8998±.0024 | 0.0840±.0014 | 0.9186±.0012 | 0.8392±.0014 |
| | LDLLC | 0.1400±.0023 | 0.4139±.0032 | 0.9045±.0109 | 0.0859±.0070 | 0.9169±.0093 | 0.8381±.0032 |
| | PT-Bayes | 0.3044±.0029 | 0.8913±.0033 | 1.9535±.0083 | 0.4238±.0054 | 0.6885±.0019 | 0.6276±.0018 |
| | OURS | **0.132±0.0053** | **0.4071±0.0156** | **0.8849±0.04** | **0.0798±0.006** | **0.9212±0.0054** | **0.8423±0.0073** |
| Nature | AA-BP | 0.4040±.0104 | 2.5361±.0126 | 7.1280±.0630 | 3.9933±.1458 | 0.5036±.0254 | 0.3678±.0175 |
| | AA-KNN | 0.3566±.0059 | 2.4758±.0096 | 6.9364±.0399 | 3.7662±.0259 | 0.6313±.0038 | 0.3948±.0045 |
| | CPNN | 0.3818±.0064 | 2.5189±.0106 | 7.1490±.0474 | 4.1641±.0636 | 0.5343±.0166 | 0.3388±.0087 |
| | LDSVR | 0.3642±.0061 | 2.4766±.0074 | 6.9645±.0277 | 3.9580±.0179 | 0.5798±.0032 | 0.3679±.0028 |
| | LCLR | **0.3583±.0036** | **2.4808±.0155** | **6.8452±.0285** | 1.1300±.0068 | 0.6766±.0047 | 0.4662±.0055 |
| | LDLSF | 0.3744±.0090 | 2.5772±.0234 | 7.1752±.0513 | 1.7217±.0090 | 0.6083±.0101 | 0.4530±.0100 |
| | LDLLC | 0.6816±.0027 | 2.8773±.0068 | 8.4701±.0160 | 2.8957±.0021 | 0.4247±.0022 | 0.3069±.0028 |
| | PT-Bayes | 0.4285±.0076 | 2.5436±.0073 | 7.2272±.0297 | 3.9983±.0667 | 0.5423±.0064 | 0.3399±.0052 |
| | OURS | 0.3714±0.015 | 2.4881±0.0229 | 6.8807±0.1052 | **0.9923±0.041** | 0.6452±0.0149 | **0.4531±0.0166** |
| fbp5500 | AA-BP | 0.1829±.0014 | 1.3551±.0014 | 2.4861±.0014 | 0.0988±.0017 | 0.8272±.0658 | 0.6421±.0859 |
| | AA-KNN | 0.3295±.0759 | 1.4446±.0014 | 2.7604±.0014 | **0.0981±.0016** | 0.7862±.0499 | 0.5884±.0376 |
| | CPNN | 0.3968£.0014 | 1.5044±.0940 | 2.9635±.0014 | 0.1819±.0014 | 0.6585±.0754 | 0.5022±.0745 |
| | LDSVR | 0.3270±.0014 | 1.4421±.0812 | 2.7538±.2398 | 0.0900±.0014 | 0.7922±.0612 | 0.5905±.0609 |
| | LCLR | 0.3377±.0180 | 1.4497±.0167 | 2.7787±.0556 | 0.5183±.0558 | 0.7805±.0373 | 0.5810±.0240 |
| | LDLSF | 0.3326±.0026 | 1.4497±.0689 | 2.7798±.0014 | 0.5184±.0027 | 0.7854±.2293 | 0.5861±.3806 |
| | LDLLC | 0.3334±.0016 | 1.4497±.0018 | 2.7808±.0053 | 0.5149±.0024 | 0.7832±.0012 | 0.5820±.0012 |
| | PT-Bayes | 0.3424±.0848 | 1.5953±.0014 | 3.3559±.0014 | 0.3005±0030 | 0.6586±.0747 | 0.4508±.0475 |
| | OURS | **0.1386±0.0033** | **1.2788±0.0079** | **2.185±0.0241** | 0.1125±0.0041 | **0.9516±0.0022** | **0.8452±0.0034** |
| RAFML | AA-BP | 0.4080±.0104 | 1.6892±.0072 | 3.7083±.0146 | 0.2950±.0014 | 0.5491±.0010 | 0.4600±.0019 |
| | AA-KNN | 0.3573±.0021 | 1.5698±.0011 | 3.3801±.0036 | **0.0803±.0458** | 0.7137±.0016 | 0.5421±.0021 |
| | CPNN | 0.4000±.0216 | **1.1745±.0338** | **2.3595±.0463** | 0.5378±0056 | 0.6301±.0047 | 0.6000±.0216 |
| | LDSVR | 0.4733±.0015 | 2.1164±.0009 | 4.9538±.0125 | 0.0760±.0009 | 0.5561±.0002 | 0.3334±.0010 |
| | LCLR | 0.3454±.0017 | 1.5577±.0050 | 3.3432±.0131 | 0.5786±.0047 | 0.7391±.0022 | 0.5550±.0020 |
| | LDLSF | 0.3477±.0016 | 1.6051±.0036 | 3.3787±.0102 | 0.5882±.0048 | 0.7335±.0029 | 0.5511±.0021 |
| | LDLLC | 0.4984±.0028 | 1.6526±.0021 | 3.4142±.0074 | 0.5834±.0024 | 0.6587±.0027 | 0.4490±.0022 |
| | PT-Bayes | 0.5971±.0035 | 2.1911±.0857 | 5.2174±.0362 | 0.8998±.0105 | 0.7248±.0113 | 0.4029±.0236 |
| | OURS | **0.1636±0.0024** | 1.4187±0.0139 | 2.7465±0.0285 | 0.2049±0.0119 | **0.9245±0.0037** | **0.7969±0.0022** |
| Movie | AA-BP | 0.1743±.0024 | 0.7322±.0036 | 1.3920±.0079 | 0.4005±.0069 | 0.8671±.0021 | 0.7569±.0017 |
| | AA-KNN | 0.1695±.0007 | 0.7123±.0022 | 1.3483±.0045 | 0.3992±.0018 | 0.8775±.0007 | 0.7596±.0008 |
| | CPNN | 0.1786±.0030 | 0.7414±.0088 | 1.4142±.0182 | 0.4299±.0108 | 0.8629±.0053 | 0.7439±.0046 |
| | LDSVR | 0.1807±.0016 | 0.7508±.0035 | 1.4321±.0077 | 0.4411±.0038 | 0.8593±.0020 | 0.7398±.0018 |
| | LCLR | 0.1654±.0042 | 0.7093±.0135 | 1.3432±.0342 | 0.1683±.0023 | 0.8827±.0026 | 0.7615±.0044 |
| | LDLSF | 0.1735±.0028 | 0.7322±.0076 | 1.3915±.0211 | 0.1829±.0030 | 0.8704±.0019 | 0.7501±.0021 |
| | LDLLC | 0.4285±.0033 | 0.7552±.0063 | 1.4398±.0174 | 0.1976±.0135 | 0.8581±.0017 | 0.7386±.0023 |
| | PT-Bayes | 0.1850±.0011 | 0.7627±.0025 | 1.4609±.0061 | 0.4506±.0025 | 0.8564±.0009 | 0.7357±.0011 |
| | OURS | **0.112±0.0044** | **0.5052±0.0146** | **0.964±0.0288** | **0.094±0.0107** | **0.9376±0.0058** | **0.8415±0.0058** |

Table 4: Comparison results (mean±std) measured by six evaluation metrics.

- LDSVR Geng & Hou (2015): LDSVR is to simultaneously fit a sigmoid function using a multi-output support vector machine.

Finally, for the mapping from the hidden layer to the output layer, we do not learn it via a Markov process but treat it as a predefined mapping based on the interval division. This configuration is shown in Table 7. Based on the above results, the following observations can be drawn:

- In most cases, our algorithm outperforms SOTA LDL methods, demonstrating that the introduction of well-structured hidden layers enables more effective feature learning and strengthens the model's representational capacity.

- The results of using the weighted allocation method are superior to the average allocation method, illustrating that the introduction of a hidden layer enhances the model's ability to capture category occurrence probabilities which improves its inference accuracy..

- Learning the transfer matrix through a Markov process achieves superior results compared to using fixed prior knowledge. We hypothesize that directly specifying the mapping prevents the incorporation of new patterns from data; moreover, if the hidden layer produces inaccurate predictions, errors may propagate and degrade output layer performance. In contrast, allowing the model to learn the internal structure adaptively, rather than imposing fixed confidence intervals, enables self-adjustment of latent representations and improves generalization.

### 4.4 TRANSFER TBLDL TO TRANDITIONAL LDL

The experiments in this section correspond to Section 3.4, where traditional LDL methods are used instead of an MLP to model the mapping from features to outputs. Due to the slow convergence of traditional iterative approaches when handling larger output spaces, we select datasets of smaller scale. Specifically, six small datasets (1–6), two medium-sized datasets (7–10), and one large dataset (12) are used. We adopt three classic LDL algorithms proposed by Geng Geng (2016), each employing the parameter settings recommended in the original literature. The comparative results are presented in Table 8. We provide the following analysis of the outcomes:

- Incorporating our framework into traditional LDL algorithms improves performance across more than 97% of the evaluation metrics, indicating that the prior category probabilities introduced by the hidden layer are beneficial for model inference.

- Our method consistently achieves lower KL divergence, a metric that measures distributional similarity, indicating that the inclusion of hidden layers smooths the predicted distributions and facilitates deeper model learning.

- Applying the proposed framework to all three traditional SOTA LDL methods results in performance improvements, demonstrating its generality and extensibility.

Compared with traditional LDL methods, although introducing hidden layers increases the output space and incurs additional computational costs, it introduces class prior probabilities and improves prediction accuracy. This performance improvement is independent of the choice of specific LDL method, demonstrating the strong generalization and transferability of our approach.

## 5 CONCLUSIONS

This paper investigates, for the first time, problems supervised with two label distributions. Specifically, we enable the model to learn the relationship between the two label distributions automatically, rather than providing the relationship a priori. To extend the applicability of our method, we generate a smoother layer distribution for problems that have only one label distribution, which contains category probability knowledge, and employ traditional LDL methods as the mapper instead of an MLP. Extensive experiments demonstrate that our approach effectively improves performance.

Our method warrants further investigation, particularly in how the transfer matrix is learned and how hidden layer label distributions are generated. Experimental results support continued development: despite being preliminary, our method demonstrates strong generalization performance and scalability, highlighting its broad potential in the LDL field.

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

## A  APPENDIX

Appendix below lists the remaining tables in the main text.

| Class | Criterion | Training | | Testing | |
|---|---|---|---|---|---|
| | | Image | Lesion | Image | Lesion |
| Mild | $1 \sim 5$ | 410 | 858 | 103 | 221 |
| Moderate | $6 \sim 20$ | 506 | 4,547 | 127 | 1,123 |
| Severe | $21 \sim 50$ | 146 | 3,857 | 36 | 890 |
| Very severe | >50 | 103 | 5,965 | 26 | 1,522 |
| Total | | 1,165 | 15,227 | 292 | 3,756 |

Table 5: Introduction of ACNE04 dataset

| | chebyshev | | clark | |
|---|---|---|---|---|
| RAFML | 0.1636±0.0024 | **0.1636±0.0021** | 1.4188±0.0143 | **1.4187±0.0139** |
| Yeast_alpha | 0.0134±0.0005 | **0.0134±0.0005** | 0.2115±0.003 | **0.2111±0.0036** |
| Yeast_cdc | 0.0162±0.0004 | **0.0162±0.0004** | **0.217±0.0055** | 0.2175±0.0054 |
| SJAFFE | **0.1234±0.0086** | 0.1235±0.0085 | 0.4374±0.0173 | **0.4373±0.0169** |
| SBU_3DFE | 0.132±0.0053 | **0.1318±0.0053** | **0.4071±0.0156** | 0.4076±0.0159 |
| Natural_Scene | 0.3714±0.015 | **0.3712±0.0149** | 2.4881±0.0229 | **2.487±0.0203** |
| fbp5500 | 0.1386±0.0033 | **0.1398±0.0029** | 1.2788±0.0079 | **1.2732±0.007** |
| Movie | 0.112±0.0044 | **0.1119±0.0046** | 0.5052±0.0146 | **0.5051±0.0136** |
| | canberra | | kldist | |
| RAFML | 2.7469±0.0302 | **2.7465±0.0285** | 0.2049±0.0126 | **0.2049±0.0119** |
| Yeast_alpha | 0.686±0.0097 | **0.6829±0.0099** | 0.0019±0.0002 | **0.0019±0.0002** |
| Yeast_cdc | 0.6496±0.0214 | **0.6511±0.0173** | 0.0041±0.0004 | **0.0041±0.0004** |
| SJAFFE | 0.9199±0.0538 | **0.9194±0.0535** | 0.0761±0.0095 | **0.0761±0.0094** |
| SBU_3DFE | **0.8849±0.04** | 0.8864±0.041 | 0.0798±0.006 | **0.0798±0.006** |
| Natural_Scene | 6.8807±0.1052 | **6.8743±0.1044** | **0.9923±0.041** | 0.9972±0.041 |
| fbp5500 | 2.185±0.0241 | **2.1723±0.0213** | **0.1125±0.0041** | 0.114±0.0049 |
| Movie | 0.965±0.0288 | **0.9645±0.0279** | 0.0941±0.0107 | **0.0941±0.0106** |
| | cosine | | intersection | |
| RAFML | 0.9244±0.0044 | **0.9245±0.0037** | 0.7969±0.0029 | **0.7969±0.0022** |
| Yeast_alpha | 0.9946±0.0002 | **0.9946±0.0002** | 0.9639±0.0006 | **0.9641±0.0005** |
| Yeast_cdc | **0.9932±0.0004** | 0.9932±0.0004 | 0.9588±0.0014 | **0.9588±0.0012** |
| SJAFFE | 0.9276±0.0084 | **0.9276±0.0083** | 0.844±0.0105 | **0.8441±0.0104** |
| SBU_3DFE | **0.9212±0.0054** | 0.9212±0.0055 | **0.8423±0.0073** | 0.842±0.0073 |
| Natural_Scene | 0.6452±0.0149 | **0.6461±0.0142** | 0.4531±0.0166 | **0.4545±0.016** |
| fbp5500 | **0.9516±0.0022** | 0.9511±0.003 | **0.8452±0.0034** | 0.8438±0.0033 |
| Movie | 0.9375±0.0058 | **0.9375±0.0057** | 0.8414±0.0058 | **0.8414±0.0058** |

Table 6: Comparison results (mean ± std) between the uniform distribution method and the weighted distribution method. The left represents the uniform distribution method, and the right represents the weighted distribution method.

| | chebyshev | | clark | |
|---|---|---|---|---|
| RAFML | 0.1648±0.0028 | **0.1636±0.0021** | 1.4198±0.0143 | **1.4187±0.0139** |
| Yeast_alpha | 0.0135±0.0004 | **0.0134±0.0005** | 0.2106±0.0028 | 0.2111±0.0036 |
| Yeast_cdc | 0.0163±0.0003 | **0.0162±0.0004** | 0.2176±0.0049 | **0.2175±0.0054** |
| SJAFFE | **0.1229±0.0088** | 0.1235±0.0085 | **0.4357±0.0175** | 0.4373±0.0169 |
| SBU_3DFE | 0.1333±0.0054 | **0.1318±0.0053** | 0.4105±0.015 | **0.4076±0.0159** |
| Natural_Scene | 0.3717±0.0145 | **0.3712±0.0149** | 2.489±0.0223 | **2.487±0.0203** |
| fbp5500 | 0.1398±0.0033 | **0.1398±0.0029** | 1.2788±0.0092 | **1.2732±0.007** |
| Movie | 0.1123±0.0047 | **0.1119±0.0046** | 0.5074±0.0171 | **0.5051±0.0136** |
| | canberra | | kldist | |
| RAFML | 2.75±0.0322 | **2.7465±0.0285** | 0.2071±0.0113 | **0.2049±0.0119** |
| Yeast_alpha | 0.6837±0.0082 | **0.6829±0.0099** | 0.0019±0.0002 | **0.0019±0.0002** |
| Yeast_cdc | 0.6518±0.0195 | **0.6511±0.0173** | 0.0041±0.0004 | **0.0041±0.0004** |
| SJAFFE | **0.9167±0.0498** | 0.9194±0.0535 | **0.0758±0.0096** | 0.0761±0.0094 |
| SBU_3DFE | 0.8917±0.0391 | **0.8864±0.041** | 0.08±0.0059 | **0.0798±0.006** |
| Natural_Scene | 6.8845±0.1012 | **6.8743±0.1044** | 0.9901±0.0385 | 0.9972±0.041 |
| fbp5500 | 2.1889±0.0261 | **2.1723±0.0213** | 0.1126±0.0049 | 0.114±0.0049 |
| Movie | 0.9674±0.0319 | **0.9645±0.0279** | 0.0942±0.0108 | **0.0941±0.0106** |
| | cosine | | intersection | |
| RAFML | 0.6604±0.0044 | **0.9245±0.0037** | 0.5111±0.0016 | **0.7969±0.0022** |
| Yeast_alpha | 0.9945±0.0001 | **0.9946±0.0002** | 0.9635±0.0005 | **0.9641±0.0005** |
| Yeast_cdc | 0.9928±0.0004 | **0.9932±0.0004** | 0.9569±0.0013 | **0.9588±0.0012** |
| SJAFFE | **0.9281±0.0085** | 0.9276±0.0083 | 0.8440±0.0099 | **0.8441±0.0104** |
| SBU_3DFE | 0.9209±0.0053 | **0.9212±0.0055** | 0.8411±0.007 | **0.842±0.0073** |
| Natural_Scene | 0.6424±0.0146 | **0.6461±0.0142** | 0.4459±0.0164 | **0.4545±0.016** |
| fbp5500 | 0.8182±0.0129 | **0.9511±0.003** | 0.6555±0.0086 | **0.8438±0.0033** |
| Movie | 0.9189±0.0055 | **0.9375±0.0057** | 0.8128±0.0061 | **0.8414±0.0058** |

Table 7: Comparison results (mean±std) by whether using Markov method to train transfer matrix. The results in the left column correspond to methods that use an a priori fixed mapping, while those in the right column are obtained by learning the transfer matrix through a Markov process.

**iislld**

| | chebyshev | | clark | | canberra | |
|---|---|---|---|---|---|---|
| Yeast_alpha | 0.0201±0.0004 | **0.0145±0.0016** | 0.3±0.0053 | **0.2117±0.0018** | 1.0036±0.0184 | **0.6651±0.0347** |
| Yeast_cdc | 0.0236±0.0009 | **0.0161±0.0003** | 0.2964±0.0098 | **0.2149±0.005** | 0.9052±0.0348 | **0.6446±0.0157** |
| Yeast_dtt | 0.0486±0.0014 | **0.0372±0.001** | 0.1296±0.004 | **0.1016±0.0024** | 0.2253±0.0071 | **0.1756±0.004** |
| Yeast_heat | 0.0531±0.0024 | **0.0432±0.0007** | 0.2261±0.0082 | **0.1881±0.0035** | 0.4581±0.0155 | **0.3745±0.0039** |
| Yeast_spo | 0.0657±0.0017 | **0.0583±0.001** | 0.2802±0.0069 | **0.2497±0.0037** | 0.5771±0.0128 | **0.5123±0.0101** |
| Yeast_spo5 | 0.0659±0.002 | **0.0587±0.0037** | 0.2793±0.007 | **0.252±0.0106** | 0.5738±0.0155 | **0.5169±0.0188** |
| SJAFFE | 0.1225±0.0034 | **0.1152±0.0127** | 0.429±0.0224 | **0.4237±0.0139** | 0.8994±0.06 | **0.875±0.0388** |
| SBU_3DFE | 0.1371±0.0036 | **0.1321±0.004** | 0.4182±0.0071 | **0.4096±0.012** | 0.9061±0.0192 | **0.8875±0.0296** |
| Movie | 0.1485±0.0028 | **0.1159±0.0037** | 0.5871±0.0122 | **0.526±0.0131** | 1.1293±0.0247 | **1.0082±0.027** |

| | kldist | | cosine | | intersection | |
|---|---|---|---|---|---|---|
| Yeast_alpha | 0.0113±0.0004 | **0.006±0.0008** | 0.9883±0.0004 | **0.9941±0.0007** | 0.9434±0.0011 | **0.9606±0.0028** |
| Yeast_cdc | 0.0131±0.0009 | **0.0069±0.0003** | 0.9868±0.001 | **0.9934±0.0003** | 0.9393±0.0024 | **0.9576±0.001** |
| Yeast_dtt | 0.0108±0.0005 | **0.0066±0.0004** | 0.9895±0.0004 | **0.9937±0.0003** | 0.9438±0.0018 | **0.9566±0.001** |
| Yeast_heat | 0.0194±0.0015 | **0.0133±0.0003** | 0.9811±0.0015 | **0.9874±0.0002** | 0.9238±0.0028 | **0.9386±0.0007** |
| Yeast_spo | 0.0303±0.0013 | **0.0244±0.0009** | 0.9712±0.0011 | **0.9771±0.0009** | 0.9044±0.0022 | **0.9157±0.0018** |
| Yeast_spo5 | 0.0302±0.0018 | **0.0248±0.002** | 0.9713±0.0017 | **0.9767±0.0019** | 0.9048±0.0027 | **0.9149±0.0032** |
| SJAFFE | 0.0747±0.0061 | **0.07±0.0096** | 0.9294±0.0048 | **0.9344±0.0092** | 0.8462±0.0094 | **0.8522±0.0089** |
| SBU_3DFE | 0.0842±0.0026 | **0.0801±0.0045** | 0.9181±0.0024 | **0.9219±0.0039** | 0.8375±0.0036 | **0.8413±0.0052** |
| Movie | 0.1338±0.006 | **0.1008±0.0058** | 0.9067±0.004 | **0.9338±0.0037** | 0.8013±0.0044 | **0.8333±0.005** |

**ptbayes**

| | chebyshev | | clark | | canberra | |
|---|---|---|---|---|---|---|
| Yeast_alpha | 0.1078±0.0043 | **0.0432±0.0082** | 1.2409±0.0491 | **0.5709±0.0632** | 4.4664±0.1927 | **1.9881±0.2408** |
| Yeast_cdc | 0.1098±0.0083 | **0.0477±0.0071** | 1.0993±0.0587 | **0.5295±0.0497** | 3.6011±0.1896 | **1.6826±0.166** |
| Yeast_dtt | 0.1749±0.0151 | **0.0922±0.0092** | 0.4747±0.0416 | **0.2493±0.0224** | 0.833±0.0735 | **0.4333±0.0384** |
| Yeast_heat | 0.171±0.0197 | **0.0969±0.0043** | 0.6836±0.0528 | **0.3915±0.0277** | 1.4454±0.1189 | **0.8061±0.0548** |
| Yeast_spo | 0.1791±0.0117 | **0.1003±0.007** | 0.6864±0.0421 | **0.4135±0.0214** | 1.453±0.0985 | **0.8619±0.052** |
| Yeast_spo5 | 0.1796±0.0267 | **0.0989±0.0033** | 0.7077±0.0781 | **0.4067±0.0109** | 1.4981±0.1853 | **0.8447±0.0247** |
| SJAFFE | 0.1222±0.0117 | **0.1203±0.0086** | **0.4297±0.0306** | 0.4320±0.0314 | 0.9089±0.0649 | **0.9032±0.0632** |
| SBU_3DFE | 0.1382±0.0025 | **0.1382±0.0019** | 0.4121±0.0063 | **0.4112±0.003** | 0.9011±0.014 | **0.9002±0.0089** |
| Movie | 0.2±0.003 | **0.1985±0.0012** | 0.8058±0.0063 | **0.7999±0.0083** | 1.5608±0.014 | **1.5446±0.0166** |

| | kldist | | cosine | | intersection | |
|---|---|---|---|---|---|---|
| Yeast_alpha | 0.3203±0.021 | **0.0687±0.0197** | 0.8357±0.0077 | **0.9487±0.0119** | 0.7591±0.0093 | **0.8895±0.0138** |
| Yeast_cdc | 0.285±0.0263 | **0.0671±0.0134** | 0.8469±0.0093 | **0.9488±0.0094** | 0.7673±0.0117 | **0.8882±0.0115** |
| Yeast_dtt | 0.2113±0.0292 | **0.0571±0.0149** | 0.8986±0.0125 | **0.9612±0.0059** | 0.8049±0.0177 | **0.8949±0.0089** |
| Yeast_heat | 0.2778±0.046 | **0.0774±0.0073** | 0.8662±0.0168 | **0.9427±0.0038** | 0.7718±0.019 | **0.8679±0.0079** |
| Yeast_spo | 0.287±0.0465 | **0.0854±0.0178** | 0.8591±0.0107 | **0.9358±0.0063** | 0.7675±0.015 | **0.8584±0.0082** |
| Yeast_spo5 | 0.2958±0.0961 | **0.0854±0.0132** | 0.8564±0.0226 | **0.9372±0.004** | 0.7617±0.0293 | **0.8613±0.0042** |
| SJAFFE | 0.0754±0.012 | **0.0737±0.0081** | 0.9289±0.0107 | **0.9305±0.0072** | 0.8447±0.0118 | **0.8463±0.0106** |
| SBU_3DFE | 0.085±0.0021 | **0.0847±0.0021** | 0.9178±0.0019 | **0.9181±0.0018** | **0.8394±0.0026** | 0.8391±0.0018 |
| Movie | 0.7028±0.0529 | **0.5891±0.0597** | 0.8504±0.0026 | **0.851±0.0012** | 0.7233±0.0024 | **0.7254±0.0015** |

**bfgslld**

| | chebyshev | | clark | | canberra | |
|---|---|---|---|---|---|---|
| Yeast_alpha | 0.0136±0.0004 | **0.0134±0.0004** | 0.2123±0.0038 | **0.2097±0.0044** | 0.6885±0.0163 | **0.6811±0.0153** |
| Yeast_cdc | 0.0172±0.0007 | **0.0165±0.0003** | **0.2154±0.011** | 0.2193±0.0039 | **0.6465±0.0271** | 0.6576±0.0125 |
| Yeast_dtt | 0.0364±0.0014 | **0.0361±0.0018** | 0.0993±0.0038 | **0.0985±0.0049** | 0.1708±0.0057 | **0.1688±0.007** |
| Yeast_heat | 0.0423±0.002 | **0.0418±0.0023** | 0.1826±0.0073 | **0.1814±0.0095** | 0.3649±0.0111 | **0.3611±0.0184** |
| Yeast_spo | 0.0583±0.002 | **0.0578±0.0015** | 0.2481±0.0083 | **0.2462±0.0054** | 0.5085±0.0177 | **0.5054±0.0093** |
| Yeast_spo5 | 0.0914±0.0021 | **0.0907±0.0029** | 0.1844±0.0058 | **0.1829±0.007** | 0.2832±0.008 | **0.2806±0.0101** |
| SJAFFE | 0.0982±0.0069 | **0.0901±0.0106** | 0.3778±0.0281 | **0.3316±0.0171** | 0.761±0.0481 | **0.6763±0.0339** |
| SBU_3DFE | 0.1186±0.0047 | **0.1155±0.0027** | 0.3772±0.0125 | **0.3759±0.0067** | 0.7964±0.0269 | **0.7922±0.0105** |
| Movie | 0.1256±0.0023 | **0.1178±0.0018** | 0.5447±0.0073 | **0.518±0.0054** | 1.0498±0.0141 | **0.995±0.0112** |

| | kldist | | cosine | | intersection | |
|---|---|---|---|---|---|---|
| Yeast_alpha | 0.0056±0.0002 | **0.0055±0.0002** | 0.9945±0.0003 | **0.9946±0.0002** | 0.962±0.001 | **0.9624±0.0009** |
| Yeast_cdc | 0.0079±0.0005 | **0.0072±0.0003** | 0.9930±0.0005 | **0.9931±0.0002** | **0.9574±0.0017** | 0.9567±0.0008 |
| Yeast_dtt | 0.0063±0.0006 | **0.0062±0.0008** | 0.994±0.0005 | **0.9942±0.0006** | 0.9578±0.0013 | **0.9584±0.0016** |
| Yeast_heat | 0.0127±0.001 | **0.0124±0.0013** | 0.9879±0.0008 | **0.9882±0.0011** | 0.94±0.0017 | **0.9408±0.0029** |
| Yeast_spo | 0.0243±0.0015 | **0.0239±0.0015** | 0.9771±0.0015 | **0.9775±0.0013** | 0.9162±0.0026 | **0.9167±0.0015** |
| Yeast_spo5 | 0.0294±0.002 | **0.0291±0.0019** | 0.974±0.0014 | **0.9744±0.0014** | 0.9086±0.0021 | **0.9093±0.0029** |
| SJAFFE | 0.0564±0.0094 | **0.044±0.0051** | 0.9475±0.0084 | **0.9578±0.005** | 0.8725±0.0077 | **0.8855±0.0076** |
| SBU_3DFE | 0.0655±0.0043 | **0.064±0.0029** | 0.9357±0.0041 | **0.9372±0.0027** | 0.8571±0.0052 | **0.8585±0.0021** |
| Movie | 0.1157±0.0048 | **0.102±0.0031** | 0.9246±0.0031 | **0.9328±0.0019** | 0.8237±0.0032 | **0.834±0.002** |

Table 8: Comparison results (mean ± std) evaluated using six evaluation metrics across three traditional LDL methods. The left line shows the results of the traditional methods, and the right line shows the results after introducing TBLDL.

