# OpenReview forum: "Two-Branch Label Distribution Learning"
_ICLR.cc/2026/Conference — Submitted to ICLR 2026_

### Official Review · Reviewer_fBs8 · 2025-10-21

**Soundness:** 2
**Presentation:** 2
**Contribution:** 2
**Rating:** 2
**Confidence:** 4

**Summary:**

This paper introduces a novel machine learning paradigm termed TBLDL, which addresses LDL problems supervised by two distinct label distributions. The core idea is that by modeling the intrinsic correlation between different label branches, a model can fuse cross-branch information to enhance predictive performance. The authors develop framework where, for datasets with only a single label branch, an auxiliary hidden layer is algorithmically generated to recast the problem into a TBLDL task, thereby injecting useful prior knowledge. Experiments across 13 datasets shows that TBLDL consistently outperforms existing approaches.

**Strengths:**

This paper proposes a novel label distribution learning method which adaptively captures the correlation between different label branches, and the proposed method outperforms several conventional LDL methods on twelve datasets.

**Weaknesses:**

1. In real-world scenarios, the requirement for dual-branch label information is seldom met in existing datasets, which may limit the practical applicability of the method. While the authors introduce a technique to convert single-branch labels into a dual-branch format, the rationale behind this augmentation process remains unclear. For instance, it is not evident whether the synthesized second branch provides meaningful additional information. Moreover, even in cases where both types of label information are available, it is worth discussing whether the two branches convey distinct information, or if they could be used interchangeably without substantially enhancing the learning outcome.

2. The proposed approach of learning from both label distributions and counts appears to integrate existing methodologies in a relatively straightforward manner. It would benefit from a clearer articulation of its conceptual or technical novelty relative to prior work.

3. The manuscript would also benefit from further refinement in language and presentation.

4. The experimental comparisons are limited to algorithms published before 2020. To more fully validate the method's advantages, the authors should include comparisons with the numerous novel LDL algorithms developed in recent years (post-2020).

**Questions:**

1. What is the practical applicability of the proposed method, given that real-world datasets rarely satisfy the requirement for dual-branch label information?
2. Can the authors provide a stronger justification for the label augmentation method? Specifically, does the synthesized second branch label offer meaningful informational gain, or is it largely redundant?
3. How does the proposed approach significantly advance beyond existing techniques? The current integration of label distributions and counts appears straightforward; could the authors clarify the specific conceptual or technical novelty introduced?
4. Why were only pre-2020 algorithms included in the comparisons? To fairly demonstrate the method's advantages, have the authors considered more recent LDL methods?

---

### Official Review · Reviewer_yUsU · 2025-10-27

**Soundness:** 1
**Presentation:** 2
**Contribution:** 1
**Rating:** 2
**Confidence:** 5

**Summary:**

The paper primarily addresses the multi-task learning problem involving acne lesion count and severity grade, proposing a ​Two-Branch Label Distribution Learning (TBLDL)​​ method. This approach is further generalized into a ​unified TBLDL framework, based on the hypothesis that a sample can be characterized by two correlated label distributions. In the design of the predictive method, a ​hidden layer is introduced​ to capture the relationship between these two label distributions, and the final prediction results are output through a ​multi-task learning framework​. For learning problems that lack two naturally associated label distributions, a hidden layer is ​constructed artificially​ to apply the TBLDL framework. Experimental validation has been conducted on relevant datasets.

**Strengths:**

(1) The proposed concept of two-branch label distribution is relatively novel.
(2) The designed method for automatically generating the hidden layer is technically feasible.

**Weaknesses:**

​​(1) Although the concept is relatively novel, its scope of application is too narrow. The label distribution learning problem itself is already constrained by the high cost of annotation, making data difficult to acquire. The additional requirement of two-branch label distributions further limits its applicability.​
(2) ​​Even for prediction problems with two-branch label distributions, or multi-task learning problems that can be easily transformed into such a form, existing multi-task learning methods and frameworks are already capable of handling them. The method proposed in this paper merely learns the relationship between tasks through a mapping from a hidden layer to the output layer, which is insufficient to demonstrate the novelty of the approach.​
(3) For the generalized two-branch label distribution learning framework designed to handle single-label distribution prediction problems, while constructing a hidden layer from the original labels and learning the mapping from this hidden layer to the output layer technically conforms to the form of two-branch label distribution learning, it does not necessarily align with the semantic intent of having two distinct branches. In other words, this is merely a technical maneuver; the label distribution constructed for the hidden layer may not be semantically meaningful or beneficial to the final task (or could even be redundant). Its rationality and necessity are therefore questionable.
(4) The paper's description of its logical focus is insufficiently clear. While it presents an excellent solution for the specific problem of Acne Image Grading and Counting, the attempt to elevate this into a new learning problem lacks persuasiveness in terms of its writing style, framework modeling, and analysis.

**Questions:**

(1) As a new learning task, we can entirely use multi-task learning methods (or combining LDL methods) to solve this problem. So, where does the contribution of this paper lie?
(2) As a new learning task, if there is no correlation between two or multiple label distributions, what is the significance of this paper?

---

### Official Review · Reviewer_JKxT · 2025-10-31

**Soundness:** 3
**Presentation:** 2
**Contribution:** 2
**Rating:** 4
**Confidence:** 4

**Summary:**

This paper introduced the problem of two-branch label distribution learning, where an LDL model is supervised by two distinct label distributions. The proposed framework enabled the generation of an auxiliary smooth distribution, thereby improving model robustness and accuracy.

**Strengths:**

1.	The proposed method can learn the intrinsic dimensionality between the data without the dependency on fixed priori knowledge.
2.	This paper first investigated problems supervised with two label distributions.

**Weaknesses:**

1.	A figure to show the difference between the proposed two-branch method and the existing one-branch method is missing.
2.	The selection of $\sigma=3$ in section 3.2.1 is not persuasive.
3.	The notation can be simplified, for example, the footnote in Eqs. (1) and (2) can be revised to only i.
4.	The formatting of this paper is not good. For example, Table 4 is too big.
5.	This paper seems to be written in a rush, due to many formatting and typo issues.
6.	There is no hyperparameter analysis on $\sigma$ and $\lambda$.
7.	The ablation study for the multiple loss-guided components is missing.

**Questions:**

1.	Why is Eq.(7) and Eq.(9) identical?
2.	How do you determine the predefined weights of the three losses in Eq.(10)?
3.	Are there some mis-bolded in Table 4? For example, the results w.r.t. kldist.

---

### Official Review · Reviewer_RzAn · 2025-11-01

**Soundness:** 3
**Presentation:** 2
**Contribution:** 2
**Rating:** 4
**Confidence:** 3

**Summary:**

This paper introduces Two-Branch Label Distribution Learning (TBLDL), an extension of traditional Label Distribution Learning (LDL). Unlike prior LDL frameworks that rely on a single label distribution, TBLDL models two correlated label branches.
The authors propose a learnable transfer matrix optimized through a Markov process to map between branches and generalize the method to single-branch LDL by generating an auxiliary hidden layer.
Experiments on 13 datasets and several classical LDL baselines demonstrate consistent improvements.

**Strengths:**

1.	Novel problem formulation.
Introducing the two-branch label distribution learning paradigm is conceptually meaningful. The framework generalizes LDL beyond a single label distribution, allowing information sharing between semantically linked targets.
2.	Unified and extensible design.
The formulation can naturally degenerate to the single-branch case and can be integrated with existing LDL methods. This shows good generality and compatibility.
3.	Methodological clarity.
The paper provides concrete mathematical definitions for label generation, distribution transfer, and Markov-based optimization, along with an explicit algorithm outline.

**Weaknesses:**

1.	Lack of theoretical rigor in the transfer mechanism
The Markov-based optimization of the transfer matrix O is described procedurally, but not formally analyzed. How does the stochastic update relate to convergence guarantees or stationary distribution stability? Does O converge to a consistent mapping under noise or label imbalance? A theoretical discussion would strengthen this component.
2.	Ambiguity in hidden layer generation for single-branch cases
The Gaussian smoothing in Eq. (11) introduces a latent layer, but this resembles kernel density estimation rather than a learnable representation.
The authors should clarify whether the hidden layer is data-driven or purely statistical, and whether its parameters (e.g., σ, nk) are optimized jointly or manually fixed.

**Questions:**

Add convergence and complexity analyses, including asymptotic bounds or empirical iteration curves.

Visualize learned mappings O to confirm meaningful alignment between hidden and output spaces.

Compare with multitask learning or canonical correlation analysis (CCA) to highlight differences between TBLDL and standard multi-output regression.

---

### Meta-Review · Area_Chair_oPfB · 2026-01-06

**Summary:**

At first glance, the paper does not appear ready for publication in its current format. Beyond presentation issues, the proposed method has a very narrow scope and lacks solid theoretical justification. Core mechanisms, such as the transfer process and hidden-layer construction, are not clearly explained or shown to be reliable.

More importantly, although the idea itself is somewhat novel, its practical usefulness is very limited. The requirement for two-branch label distributions makes the method applicable to only a small set of problems, many of which can already be handled by existing multi-task learning methods. While the approach works for a specific task, the paper does not convincingly establish it as a broadly useful or well-founded new learning framework.

Without rebuttal, it is hard to convince the paper can be accepted.

**Reviewer Concerns:**

No rebuttal.

**Reviewer Scores:**

No idea without rebuttal.

---

### Decision · Program_Chairs · 2026-01-26

Reject